# Dissecting the Nuclear Import of the Ribosomal Protein Rps2 (uS5)

**DOI:** 10.3390/biom13071127

**Published:** 2023-07-14

**Authors:** Andreas Steiner, Sébastien Favre, Maximilian Mack, Annika Hausharter, Benjamin Pillet, Jutta Hafner, Valentin Mitterer, Dieter Kressler, Brigitte Pertschy, Ingrid Zierler

**Affiliations:** 1Institute of Molecular Biosciences, University of Graz, Humboldtstrasse 50, 8010 Graz, Austria; andreas.steiner23@yahoo.com (A.S.); maximilian.mack@uni-graz.at (M.M.); valentin.mitterer@uni-graz.at (V.M.); 2BioTechMed-Graz, Mozartgasse 12/II, 8010 Graz, Austria; 3Unit of Biochemistry, Department of Biology, University of Fribourg, Chemin du Musée 10, 1700 Fribourg, Switzerland; sebastien.favre@unifr.ch (S.F.); benjamin.pillet@unifr.ch (B.P.); dieter.kressler@unifr.ch (D.K.)

**Keywords:** ribosomal protein, ribosome assembly, nuclear import, Rps2, uS5, importin, Pse1, Kap121, dedicated chaperone, yeast

## Abstract

The ribosome is assembled in a complex process mainly taking place in the nucleus. Consequently, newly synthesized ribosomal proteins have to travel from the cytoplasm into the nucleus, where they are incorporated into nascent ribosomal subunits. In this study, we set out to investigate the mechanism mediating nuclear import of the small subunit ribosomal protein Rps2. We demonstrate that an internal region in Rps2, ranging from amino acids 76 to 145, is sufficient to target a 3xyEGFP reporter to the nucleus. The importin-β Pse1 interacts with this Rps2 region and is involved in its import, with Rps2 residues arginine 95, arginine 97, and lysine 99 being important determinants for both Pse1 binding and nuclear localization. Moreover, our data reveal a second import mechanism involving the N-terminal region of Rps2, which depends on the presence of basic residues within amino acids 10 to 28. This Rps2 segment overlaps with the binding site of the dedicated chaperone Tsr4; however, the nuclear import of Rps2 via the internal as well as the N-terminal nuclear-targeting element does not depend on Tsr4. Taken together, our study has unveiled hitherto undescribed nuclear import signals, showcasing the versatility of the mechanisms coordinating the nuclear import of ribosomal proteins.

## 1. Introduction

The ribosome is a remarkable and extremely efficient macromolecular RNA–protein machine that synthesizes all proteins in the cytoplasm of the eukaryotic cell and is composed of a small 40S and a large 60S subunit. With the help of several hundred different assembly factors, the two subunits are assembled from ribosomal RNAs (rRNAs) and ribosomal proteins (r-proteins) in a highly conserved and complex maturation pathway called ribosome biogenesis, which is best studied in the yeast *Saccharomyces cerevisiae*. This assembly and maturation process starts in the nucleolus, a non-membrane-enclosed sub-compartment of the nucleus, progresses in the nucleoplasm, and ends in the cytoplasm, where the two mature subunits form the translation-competent 80S ribosome [1,2,3,4].

R-proteins have to overcome two major obstacles before they are assembled into pre-ribosomal particles in the nucle(ol)us. First, as they contain highly basic regions as well as unstructured N- or C-terminal extensions and internal loops, which engage in interactions with the negatively charged rRNA in the ribosome, r-proteins are prone to aggregation as long as they are not assembled into (pre-)ribosomal subunits [5]. Second, since r-proteins are synthesized in the cytoplasm but most of them are incorporated into pre-ribosomes in the nucle(ol)us, they need to be imported into the nucleus through the nuclear pore complex (NPC) before becoming available for ribosome biogenesis [6].

Two classes of proteins are specialized in helping to overcome these obstacles and to ensure the safe delivery of r-proteins at their site of assembly with the rRNA: (1) Dedicated chaperones of r-proteins: More than ten r-proteins were found to require protection by these specialized chaperones [5,7,8,9,10]. Most dedicated chaperones bind r-proteins already co-translationally and protect them from aggregation, presumably through shielding positively charged surfaces [5,7,11]. (2) Importins of the karyopherin superfamily: Besides mediating the nuclear import of many diverse substrate proteins (cargoes) across the hydrophobic channel of the NPC [6], selected importins have also been shown to facilitate nuclear import of r-proteins and to prevent their aggregation [12,13].

Importins usually recognize their cargo proteins by binding to basic nuclear localization sequences (NLSs). All importin-ßs have the ability to interact with the hydrophobic FG-repeat meshwork in the central channel of the NPC. Additionally, most importin-ßs, including Pse1 (also called Kap121), Kap123, and Kap104, can recognize the NLS of their cargoes directly; only the importin-ß Kap95 requires an adaptor, the importin-α Srp1 (also called Kap60), for binding to the NLS-containing cargo protein. Importin-α recognizes classical NLS sequences, which can be either monopartite (consensus K-K/R-X-K/R; where X can be any amino acid) or bipartite (K/R-K/R-X_10–12_ K/R_3/5_; with K/R_3/5_ being five residues containing at least three Ks or Rs). The importin-ß Kap104 recognizes PY-NLSs containing an N-terminal basic (or hydrophobic) motif and a C-terminal R/K/H-X_2–5_-P-Y/L/F motif. Additionally, Kap104 can bind to RGG regions (RG-rich NLSs). Pse1 binds to IK-NLSs with the consensus K-V/I-X-K-X_1–2_-K/H/R. Importantly, however, many cargoes of the above-named importins do not harbor sequences following the so-far-described NLS consensus sequences, and, moreover, for many importins, no targeting signals have been defined at all [6].

After nuclear import, the importins release their cargo proteins by binding to the small GTPase Ran in its GTP-bound form (RanGTP), which is highly concentrated in the nucleus, thereby controlling the directionality of transport [6]. Nuclear import of r-proteins is believed to be mainly performed by the non-essential importin-β Kap123, with some redundant contribution of the essential Pse1 [13].

In contrast to that notion, studies in recent years by us and others on the coordination of chaperoning and nuclear import of r-proteins revealed that several r-proteins employ importins other than Kap123. Rps3 is imported into the nucleus as a dimer, with one N-terminal domain protected by its dedicated chaperone Yar1 and the second one bound by the Srp1/Kap95 importin-α/importin-β dimer [14,15]. Rpl5 and Rpl11 are imported in complex with their dedicated chaperone Syo1, which functions as a transport adaptor for Kap104 [16]. Rpl4 contains at least five different NLS sequences and is imported into the nucleus in complex with its dedicated chaperone Acl4 by importin Kap104 [17,18,19]. Last but not least, Rps26 can be imported into the nucleus by Kap123, Kap104, or Pse1, and is then released from the importin in a RanGTP-independent manner by its dedicated chaperone Tsr2 [20].

We are interested in the nuclear import of r-protein Rps2 (also called uS5 [21]). Rps2 has an evolutionarily conserved dedicated chaperone, Tsr4 (PDCD2 in humans), which binds co-translationally to its unstructured N-terminal extension [7,10,22]. In the absence of Tsr4, Rps2 accumulates in the nucleus, suggesting that Tsr4 is required for the efficient incorporation of Rps2 into pre-ribosomes [7]. How Rps2 is imported into the nucleus has, however, remained elusive.

In this study, we uncovered that an internal fragment of Rps2, Rps2(76–145), interacts with the importin-β Pse1 and is sufficient to target a 3xyEGFP reporter into the nucleus, indicating that it contains a functional NLS. Nuclear import of Rps2(76–145)-3xyEGFP was blocked in a *pse1-1* mutant or upon changing three basic residues in Rps2, arginine 95 (R95), R97, and lysine 99 (K99), to alanines (As), suggesting that these residues are part of the NLS. Surprisingly, when fusing a larger N-terminal Rps2 fragment, Rps2(1–145), to the 3xyEGFP reporter, nuclear targeting was no longer disturbed by the R95A, R97A, and K99A exchanges. Moreover, we identified a sequence in the N-terminal part of Rps2 (amino acids 10–28), whose basic residues are essential for the nuclear targeting of the Rps2(1–145)-3xyEGFP fusion protein bearing the R95A, R97A, and K99A mutations, strongly suggesting the presence of a second NLS in the eukaryote-specific N-terminal extension of Rps2. Our results moreover revealed that import, both via the internal and the N-terminal nuclear targeting region, also occurs in the absence of Tsr4.

## 2. Methods

### 2.1. Yeast Strains and Genetic Methods

All *S. cerevisiae* strains used in this study are listed in Appendix A. Yeast plasmids were constructed using standard recombinant DNA techniques and are listed in Appendix A. All DNA fragments amplified by PCR were verified by sequencing.

The *KAP104* shuffle strain was transformed with the YCplac22-*KAP104* or the pRS314-*kap104-16* plasmid (*TRP1*), respectively, and transformed cells were streaked on 5-FOA (Thermo Scientific) plates to counter-select against the pRS316-*KAP104* (*URA3*) shuffle plasmid. After plasmid shuffling, cells were grown on plates lacking tryptophan (SDC-trp) and transformed with the *LEU2* plasmid expressing Rps2(76–145)-3xyEGFP.

### 2.2. Fluorescence Microscopy

Yeast strains were grown at 30 °C in SDC media lacking leucine (SDC-leu) to an OD_600_ of ~0.5 (logarithmic growth phase). Cells were imaged by fluorescence microscopy using a Leica DM6 B microscope, equipped with a DFC 9000 GT camera, using the PLAN APO 100× objective, narrow-band GFP or RHOD ET filters, and LasX software. Full-length Rps2 as well as fragments and variants thereof were expressed with a C-terminal 3xyEGFP tag under the transcriptional control of the *ADH1* promoter from a centromeric *LEU2* plasmid. Plasmids expressing these 3xyEGFP reporter proteins were transformed into a Nop58-yEmCherry expressing strain, the C303 wild-type strain, *rps2*∆ and *tsr4*∆ *rps2*∆ strains containing a centromeric *URA3*-*RPS2* plasmid, or the indicated importin mutant strains. Since Rps2 is an essential protein, we investigated the localization of the different Rps2-3xyEGFP fusion proteins in strains harboring the wild-type *RPS2* gene either at the chromosomal locus or on a plasmid.

### 2.3. Yeast Two-Hybrid (Y2H) Assays

Protein–protein interactions between Rps2 (and fragments/variants thereof) and Pse1/Pse1.302C or Tsr4 were analyzed by yeast two-hybrid (Y2H) assays using the reporter strain PJ69-4A. This Y2H strain allows for the detection of both weak (*HIS3* reporter) and strong interactions (*ADE2* reporter). Two plasmids were co-transformed into PJ69-4A, whereby one plasmid was expressing fusions to the Gal4 DNA-binding domain (G4BD, BD, *TRP1* marker) and the other fusions to the Gal4 transcription activation domain (G4AD, AD, *LEU2* marker). For the Rps2-Tsr4 Y2H interaction assays, Rps2 variants, C-terminally fused to the G4BD, and full-length Tsr4, C-terminally fused to the G4AD, were expressed from centromeric (CEN, low-copy) plasmids (pG4BDC22 and pG4ADC111, respectively). For the Rps2-Pse1 Y2H interaction assays, Rps2 variants, either N- or C-terminally fused to the G4BD, and Pse1 or Pse1.302C, C-terminally fused to the G4AD, were expressed from episomal (2µ, high-copy) plasmids (pG4BDN112, pGAG4BDC112, and pGAG4ADC181, respectively).

After the selection of transformants on plates lacking leucine and tryptophan (SDC-leu-trp, -LT), cells were spotted onto SDC-leu-trp plates as well as onto plates lacking histidine, leucine, and tryptophan (SDC-his-leu-trp, -HLT), and lacking adenine, leucine, and tryptophan (SDC-ade-leu-trp, -ALT), respectively. Plates were incubated for 3 days at 30 °C.

### 2.4. Tandem Affinity Purification (TAP)

For TAP purifications, plasmids expressing Rps2(76–145) or Rps2(76–145).R_95_R_97_K_99>A_, C-terminally fused to the TAP tag, or a plasmid expressing the TAP tag alone were transformed into a haploid W303-derived wild-type strain. Cells were grown in 4 l yeast extract peptone dextrose medium (YPD) to an optical density (OD_600_) of 2 at 30 °C.

TAP purifications were performed in a buffer containing 50 mM Tris-HCl (pH 7.5), 100 mM NaCl, 1.5 mM MgCl_2_, 0.1% NP-40, 1 mM dithiothreitol (DTT), and 1x Protease Inhibitor Mix FY (Serva). Cells were lysed by mechanical disruption using glass beads and the cell lysate was incubated with 300 µL IgG Sepharose^TM^ 6 Fast Flow (GE Healthcare, Chicago, IL, USA) for 60 min at 4 °C. After incubation, the IgG Sepharose^TM^ beads were transferred into Mobicol columns (MoBiTec, Göttingen, Germany) and washed with 10 mL buffer. Then, TEV protease was added and elution from the beads was performed under rotation for 90 min at room temperature. After the addition of 2 mM CaCl_2_, TEV eluates were incubated with 300 µL Calmodulin Sepharose^TM^ 4B beads (GE Healthcare) for 60 min at 4 °C. After washing with 5 mL buffer containing 2 mM CaCl_2_, proteins were eluted from Calmodulin Sepharose^TM^ with elution buffer consisting of 10 mM Tris-HCl (pH 8.0), 5 mM EGTA, and 50 mM NaCl under rotation for 20 min at room temperature. The protein samples were separated on NuPAGE^TM^ 4–12% Bis-Tris gels (Invitrogen, Carlsbad, CA, USA) followed by Western blotting.

### 2.5. Western Blotting

Western blot analysis was performed using the following antibodies: α-CBP antibody (1:5000; Merck-Millipore, Burlington, MA, USA, cat. no. 07-482), α-Pse1 antibody (1:500; Matthias Seedorf [22]), secondary α-rabbit horseradish peroxidase-conjugated antibody (1:15,000; Sigma-Aldrich, St. Louis, MO, USA, cat. no. A0545). Protein signals were visualized using the Clarity^TM^ Western ECL Substrate Kit (Bio-Rad, Hercules, CA, USA) and captured by the ChemiDoc^TM^ Touch Imaging System (Bio-Rad).

### 2.6. TurboID-Based Proximity Labeling

Plasmids expressing C-terminal TurboID-tagged bait proteins under the control of the copper-inducible *CUP1* promoter were transformed into the wild-type strain YDK11-5A. Transformed cells were grown at 30 °C in 100 mL SDC-leu medium, prepared with copper-free yeast nitrogen base (FORMEDIUM), to an OD_600_ between 0.4 and 0.5. Then, copper sulfate, to induce expression from the *CUP1* promoter, and freshly prepared biotin (Sigma-Aldrich, St. Louis, MO, USA) were added to a final concentration of 500 μM, and cells were grown for an additional hour, typically reaching a final OD_600_ between 0.6 and 0.8, and harvested by centrifugation at 4000 rpm for 5 min at 4 °C. Then, cells were washed with 50 mL ice-cold H_2_O, resuspended in 1 mL ice-cold lysis buffer (LB: 50 mM Tris-HCl (pH 7.5), 150 mM NaCl, 1.5 mM MgCl2, 0.1% SDS, and 1% Triton X-100) containing 1 mM PMSF, transferred to 2 mL safe-lock tubes, pelleted by centrifugation, frozen in liquid nitrogen, and stored at −80 °C. Extracts were prepared, upon the resuspension of cells in 400 μL lysis buffer containing 0.5% sodium deoxycholate and 1 mM PMSF (LB-P/D), by glass bead lysis with a Precellys 24 homogenizer (Bertin Technologies, Montigny-le-Bretonneux, France) set at 5000 rpm using a 3 × 30 s lysis cycle with 30 s breaks in between at 4 °C. Lysates were transferred to 1.5 mL tubes. For complete extract recovery, 200 μL LB-P/D was added to the glass beads and, after brief vortexing, combined with the already transferred lysate. Cell lysates were clarified by centrifugation for 10 min at 13,500 rpm at 4 °C, transferred to a new 1.5 mL tube. Total protein concentration in the clarified cell extracts was determined with the Pierce™ BCA Protein Assay Kit (Thermo Scientific, Waltham, MA, USA) using a microplate reader (BioTek 800 TS). To reduce non-specific binding, 100 μL of Pierce™ High Capacity Streptavidin Agarose Resin (Thermo Scientific) slurry, corresponding to 50 μL of settled beads, were transferred to a 1.5 mL safe-lock tube, blocked by incubation with 1 mL LB containing 3% BSA for 1 h at RT, and then washed four times with 1 mL LB. For the affinity purification of biotinylated proteins, 2 mg of total protein in an adjusted volume of 800 μL LB-P/D was added to the blocked and washed streptavidin beads, and binding was carried out for 1 h at RT on a rotating wheel. Beads were then washed once for 5 min with 1 mL of wash buffer (50 mM Tris-HCl (pH 7.5), 2% SDS), five times with 1 mL LB, and finally five times with 1 mL ABC buffer (100 mM ammonium bicarbonate (pH 8.2)). Bound proteins were eluted by two consecutive incubations with 30 μL 3× SDS sample buffer, containing 10 mM biotin and 20 mM DTT, for 10 min at 75 °C. The eluates were combined in one 1.5 mL safe-lock tube and stored at –20 °C. Upon reduction with DTT and alkylation with iodoacetamide, samples were separated on NuPAGE 4–12% Bis-Tris gels (Invitrogen, Carlsbad, CA, USA), run in NuPAGE 1× MES SDS running buffer (Novex) at 200 V for a total of 12 min. The gels were incubated with Brilliant Blue G Colloidal Coomassie (Sigma-Aldrich) until the staining of proteins was visible. Each lane was cut, from slot to the migration front, into three gel pieces that were, upon their fragmentation into smaller pieces, transferred into separate 1.5 mL low-binding tubes.

Gel pieces were covered with 100–150 μL of ABC buffer, prepared in HPLC-grade H_2_O, and incubated for 10 min at RT in a thermoshaker set to 1000 rpm. Then, gel pieces were covered with 100–150 μL of HPLC-grade absolute EtOH and incubated for 10 min at RT in a thermoshaker set to 1000 rpm. These two wash steps were repeated two more times. For the in-gel digestion of proteins, gel pieces were covered with 120 μL of ABC buffer containing 1 μg sequencing-grade modified trypsin (Promega Madison, WI, USA) and incubated overnight at 37 °C with shaking at 1000 rpm. To stop the digestion and recover the peptides, 50 μL of a 2% trifluoroacetic acid (TFA) solution was added, and, after a 10 min incubation at RT with shaking at 1000 rpm, the supernatant was transferred to a new 1.5 mL low-binding tube. The gel pieces were then incubated, again for 10 min at RT with shaking at 1000 rpm, another two times with 100–150 μL EtOH, and these two supernatants were combined with the first supernatant. Finally, using a SpeedVac, the organic solvents were evaporated and the volume was reduced to around 50 μL. Then, 200 μL of buffer A (0.5% acetic acid) were added, and the samples were applied to C18 StageTips [23], equilibrated with 50 μL of buffer B (80% acetonitrile, 0.3% TFA) and washed twice with 50 μL of buffer A, for desalting and peptide purification. StageTips were washed once with 100 μL of buffer A, and the peptides were eluted with 50 μL of buffer B. The solvents were completely evaporated using a SpeedVac. Peptides were resuspended by first adding 3 µL buffer A* (3% acetonitrile, 0.3% TFA) and then 17 µL buffer A*/A (30% buffer A*/70% buffer A), with each solvent addition being followed by vortexing for 10 s. Samples were stored at –80 °C.

LC-MS/MS measurements were performed on a Q Exactive HF-X (Thermo Scientific) coupled to an EASY-nLC 1200 nanoflow-HPLC (Thermo Scientific). HPLC-column tips (fused silica) with 75 μm inner diameter were self-packed with ReproSil-Pur 120 C18-AQ, 1.9 μm particle size (Dr. Maisch GmbH, Ammerbuch, Germany) to a length of 20 cm. Samples were directly applied onto the column without a pre-column. A gradient of A (0.1% formic acid in H_2_O) and B (0.1% formic acid in 80% acetonitrile in H_2_O) with increasing organic proportion was used for peptide separation (loading of sample with 0% B; separation ramp: from 5–30% B within 85 min). The flow rate was 250 nL/min and for sample application, it was 600 nL/min. The mass spectrometer was operated in the data-dependent mode and switched automatically between MS (max. of 1 × 10^6^ ions) and MS/MS. Each MS scan was followed by a maximum of ten MS/MS scans using a normalized collision energy of 25% and a target value of 1000. Parent ions with a charge state form z = 1 and unassigned charge states were excluded for fragmentation. The mass range for MS was *m*/*z* = 370–1750. The resolution for MS was set to 70,000 and for MS/MS to 17,500. MS parameters were as follows: spray voltage 2.3 kV, no sheath and auxiliary gas flow, ion-transfer tube temperature 250 °C.

The MS raw data files were analyzed with the MaxQuant software package version 1.6.2.10 [24] for peak detection, generation of peak lists of mass-error-corrected peptides, and database searches. The UniProt *Saccharomyces cerevisiae* database (version March 2016), additionally including common contaminants, trypsin, TurboID, and GFP, was used as reference. Carbamidomethylcysteine was set as fixed modification and protein amino-terminal acetylation, oxidation of methionine, and biotin were set as variable modifications. Four missed cleavages were allowed, enzyme specificity was Trypsin/P, and the MS/MS tolerance was set to 20 ppm. Peptide lists were further used by MaxQuant to identify and relatively quantify proteins using the following parameters: peptide and protein false discovery rates, based on a forward–reverse database, were set to 0.01, minimum peptide length was set to seven, and minimum number of unique peptides for identification and quantification of proteins was set to one. The ‘match-between-run’ option (0.7 min) was used.

For quantification, missing iBAQ (intensity-based absolute quantification) values in the two control purifications from cells expressing either the GFP-TurboID or the NLS-GFP-TurboID bait were imputed in Perseus [25]. For normalization of intensities in each independent purification, iBAQ values were divided by the median iBAQ value, derived from all nonzero values, of the respective purification. To calculate the enrichment of a given protein compared to its average abundance in the two control purifications, the normalized iBAQ values were log2-transformed and those of the control purifications were subtracted from the ones of each respective bait purification. For graphical presentation, the normalized iBAQ value (log10 scale) of each protein detected in a given bait purification was plotted against its relative abundance (log2-transformed enrichment compared to the control purifications). To visualize the effects of the RRK>A mutations on the proximal protein neighborhood of Rps2, the normalized iBAQ value (log10 scale) of each protein detected in the purification from cells expressing a wild-type Rps2 bait protein (full-length Rps2, Rps2(1–145), or Rps2(76–145)) was plotted against its relative abundance (log2-transformed enrichment) compared to the purification from cells expressing the respective RRK>A mutant protein (with prior imputation of missing iBAQ values).

## 3. Results

### 3.1. Rps2 Amino Acids 76–145 Are Sufficient to Target the Protein to the Nucleus

Like most other r-proteins, Rps2 assembles into pre-ribosomal particles in the nucleus [26], necessitating nuclear import of newly synthesized Rps2. Yeast Rps2 is recognized co-translationally by its dedicated chaperone Tsr4 in the cytoplasm [7,10]; however, the mechanism by which Rps2 is imported into the nucleus has so far remained elusive.

We first aimed to narrow down the part of Rps2 that is capable of targeting the protein to the nucleus. Since full-length Rps2 is imported into the nucleus, incorporated into pre-ribosomal particles, and subsequently, as a component of these, exported to the cytoplasm, the majority of all cellular Rps2 is present in cytoplasmic 40S subunits. We reasoned, however, that small sub-fragments of Rps2 would most likely not become incorporated into pre-ribosomes and could hence be visualized in the nucleus in case they carry an NLS. We designed a series of Rps2 fragments with overlapping regions, and constructed plasmids encoding these Rps2 fragments (Figure 1A, Appendix A), each fused to a 3xyEGFP tag at the C-terminus. In order to differentiate between different types of nuclear localization, we utilized a strain expressing the nucleolar marker protein Nop58 fused to mCherry (Nop58-yEmCherry) from the genomic locus. A localization of 3xyEGFP reporter fusion proteins exclusively in the nucleolus, the site where ribosome biogenesis starts, would result in a perfect overlap with the Nop58-yEmCherry signal. Conversely, a nucleoplasmic localization of the 3xyEGFP reporter fusions would result in a signal adjacent to the Nop58-yEmCherry nucleolar signal with no overlap. Finally, reporter fusions localizing to both nuclear subcompartments would exhibit a larger oval-shaped GFP signal and partially overlap with the Nop58-yEmCherry marker.

We transformed the Nop58-yEmCherry-expressing strain with plasmids encoding the Rps2 fragment 3xyEGFP fusions, and inspected the localization of the reporter proteins by fluorescence microscopy (Figure 1B). As expected, full-length Rps2-3xyEGFP showed a predominantly cytoplasmic signal. Additionally, we occasionally observed, as previously described [7], small dot-like structures that likely correspond to aggregates. The occurrence and size of these dot-like structures was strongly increased for several of the 3xyEGFP-fused Rps2 fragments, particularly for Rps2(118–218) and to a lesser extent also for Rps2(23–75) and Rps2(175–254). Moreover, the localization of Rps2(118–218)-3xyEGFP might correspond to a mitochondrial staining. We speculate that the above Rps2 fragments, as they are no longer embedded in the full-length protein context, are especially prone to misfolding, which may lead to their aberrant localization and/or increase their aggregation. Among these, Rps2(23–75)3xyEGFP localized to the entire nucleus and the cytoplasm, with a stronger signal in the nucleus. Last but not least, we identified two 3xyEGFP-fused Rps2 fragments that did not exhibit, when compared to full-length Rps2, increased aggregate formation: Rps2(1–42)-3xyEGFP localized both to the cytoplasm and the entire nucleus, with a slightly stronger signal in the nucleus. More strikingly, Rps2(76–145)-3xyEGFP localized exclusively to the nucleus, suggesting that this Rps2 fragment is sufficient to target the 3xyEGFP reporter to the nucleus (Figure 1B). The nuclear Rps2(76–145)-3xyEGFP signal appeared weaker in the area overlapping with Nop58-yEmCherry compared to the rest of the nucleus. To conclude, our data suggest that the Rps2(76–145)-3xyEGFP fragment contains a functional NLS, which mediates the targeting of this fragment to the nucleoplasm.

### 3.2. Rps2 Residues R95, R97, and K99 Are Essential for Nuclear Targeting of Rps2(76–145)-3xyEGFP

As attempts to narrow down the sequence responsible for nuclear targeting by further N- or C-terminal truncation of the Rps2(76–145) fragment resulted in mitochondrial staining or increased aggregate formation, respectively, suggesting misfolding of the resulting 3xyEGFP fusion proteins, we instead used the Rps2(76–145)-3xyEGFP fusion as a starting point to introduce the selected amino acid exchanges into potential NLS segments.

As the sequence of the Rps2(76–145) fragment does not contain any obvious so-far-described NLS, we searched for clusters of basic amino acids that are conserved in eukaryotic Rps2. We considered Rps2 residues 95 to 99 (95-RTRFK-99), notably containing three basic amino acids, as a candidate sequence that might contribute to a non-classical NLS (Figure 2A, Appendix A).

To address whether R95, R97, and K99 indeed contribute to the nuclear import of the Rps2(76–145) fragment, we constructed a plasmid expressing the Rps2(76–145).R_95_R_97_K_99>A_-3xyEGFP variant (abbreviated as 76–145 RRK>A in Figures) in which all three basic amino acids were exchanged to As. Next, we compared the localization of Rps2(76–145).R_95_R_97_K_99>A_-3xyEGFP with the one of Rps2(76–145)-3xyEGFP in the yeast strain expressing Nop58-yEmCherry by fluorescence microscopy (Figure 2B, Appendix A). Indeed, the R95A, R97A, and K99A exchanges (95-ATAFA-99 sequence instead of 95-RTRFK-99) resulted in an almost complete shift of the otherwise nuclear Rps2(76–145)-3xyEGFP fragment to the cytoplasm. We conclude that Rps2 contains a functional NLS within amino acids 76–145, with residues R95, R97, and K99 being essential features of this NLS. Importantly, considering that Tsr4 interacts with the very N-terminal part of Rps2 [7], which is not present in the tested Rps2(76–145) fragment (Figure 2A), import via this sequence has to be independent of Tsr4.

### 3.3. Import of Rps2(76–145) Is Mediated by Pse1

To gain better insight into the nuclear import of Rps2 mediated by amino acids 76–145, we aimed to identify the importin(s) that recognize the novel Rps2 NLS. To this end, we analyzed the localization of the Rps2(76–145)-3xyEGFP reporter in different importin mutant strains (Figure 3A, Appendix A). The nuclear localization of the Rps2(76–145)-3xyEGFP fusion protein remained largely unaffected in *srp1-31*, *kap95*-ts, and *kap104*-*16* importin mutants (Appendix A). Rps2(76–145)-3xyEGFP appeared to show a reduced nuclear localization in some cells in the *kap123*∆ strain (Appendix A). To better distinguish whether or not *kap123*∆ cells have a slight Rps2(76–145)-3xyEGFP import defect, we transformed the cells with a plasmid containing the *KAP123* wild-type gene and assessed whether this would result in an increased nuclear signal, which would be an indication for complementation of a potential import defect. As reported before [22], *kap123*∆ cells did not display any growth defects, and as expected, growth was unaltered upon transformation of the *KAP123*-containing plasmid (Appendix A). Moreover, Rps2(76–145)-3xyEGFP displayed a similar localization in *kap123*∆ cells either transformed with *KAP123*-containing or empty plasmid; hence, no complementation of a potential defect was observed (Appendix A). We conclude that *kap123*∆ cells do not show an Rps2(76–145)-3xyEGFP import defect.

Last but not least, a strong reduction in nuclear accumulation of Rps2(76–145)-3xyEGFP was observed in *pse1-1* mutant cells (Figure 3A). Importantly, transformation of the *pse1-1* mutant with a plasmid containing the *PSE1* wild-type gene complemented the growth defect of *pse1-1* mutant cells (Appendix A), as well as their defect in the nuclear import of the Rps2(76–145)-3xyEGFP reporter protein (Figure 3B). Altogether, our data suggest that the NLS within Rps2(76–145) is mainly recognized by the importin-β Pse1.

To further confirm the interaction between Pse1 and Rps2(76–145) and to address whether R95, R97, and K99 are required for this interaction, we performed tandem affinity purification of C-terminally TAP-tagged Rps2(76–145) and Rps2(76–145).R_95_R_97_K_99>A_, both expressed from plasmid in a wild-type strain, and compared the extent of Pse1 co-purification (Figure 3C). As expected, Pse1 co-purified with Rps2(76–145)-TAP in a two-step affinity purification, but not with the TAP tag alone (-). Pse1 was also co-purified with Rps2(76–145).R_95_R_97_K_99>A_-TAP; however, the enrichment of Pse1 relative to the amounts of the purified bait was clearly less pronounced in the case of the Rps2(76–145).R_95_R_97_K_99>A_ protein.

To obtain additional evidence for a preferential binding of Pse1 to wild-type Rps2(76–145) in vivo, we performed TurboID-based proximity labeling to identify the proteins that are in physical proximity of C-terminally TurboID-tagged Rps2(76–145) and Rps2(76–145).R_95_R_97_K_99>A_, both expressed from plasmid under the transcriptional control of the copper-inducible *CUP1* promoter (Figure 3D, Appendix A (panels in first row), Appendix A). Indeed, Pse1 was among the most strongly enriched proteins in the affinity purification of biotinylated proteins from cells expressing wild-type Rps2(76–145), while Pse1 was not enriched when the TurboID experiment was performed with the Rps2(76–145).R_95_R_97_K_99>A_ mutant protein. These results suggest that the R95, R97, and K99 exchanges in Rps2(76–145) reduce the binding of Pse1. The reduced binding of Pse1 is most likely the reason for the nuclear import defect observed in *pse1-1* mutant cells. We conclude that Pse1 drives the nuclear import of Rps2(76–145), with R95, R97, and K99 being important determinants for full Pse1 binding.

### 3.4. Rps2 Contains a Second NLS in Its N-Terminal Extension

Having established that Rps2 contains a functional NLS in an internal region of the r-protein, we went on to test the localization of an Rps2 fragment containing both the NLS and the Tsr4-binding site and constructed a reporter plasmid for the expression of Rps2(1–145)-3xyEGFP (Figure 2A). Fluorescence microscopy of a wild-type strain transformed with the plasmid revealed that Rps2(1–145)-3xyEGFP was, similarly to Rps2(76–145)-3xEGFP, localized in the nucleus (Figure 4A). Next, we assessed the localization of the Rps2(1–145)-3xyEGFP fusion protein additionally carrying the R_95_R_97_K_99_>A exchanges in the wild-type strain. Strikingly, in contrast to the strong shift to the cytoplasm of the Rps2(76–145).R_95_R_97_K_99_>A-3xEGFP reporter, Rps2(1–145).R_95_R_97_K_99_>A-3xEGFP was still mainly found in the nucleus (Figure 4A, Appendix A), suggesting that an element within amino acids 1–75 of Rps2 ensures the nuclear targeting of Rps2(1–145), even when the above-identified Pse1-dependent NLS is rendered non-functional by the mutation of residues R95, R97, and K99.

We also investigated the localization of the above Rps2-3xyEGFP reporter fusions in the *pse1-1* mutant strain. As described above (Figure 3A), Rps2(76–145)-3xEGFP was shifted to the cytoplasm in the *pse1-1* mutant strain compared to the wild-type strain (Figure 4A). As expected, also the Rps2(76–145).R_95_R_97_K_99_>A-3xEGFP reporter showed a predominantly cytoplasmic signal in the *pse1-1* mutant, similar to the wild-type strain. The Rps2(1–145)-3xyEGFP reporter, although still showing the highest signal intensity in the nucleus, was slightly shifted to the cytoplasm, as opposed to the exclusively nuclear signal of the same fragment in the wild-type strain (Figure 4A). An even stronger shift to the cytoplasm was observed for the Rps2(1–145).R_95_R_97_K_99_>A-3xEGFP reporter in the *pse1-1* strain (Figure 4A). These results indicate that Pse1 contributes to the nuclear import of the Rps2(1–145) fragment even when R95, R97, and K99 are mutated, suggesting that there has to be a second Pse1-dependent NLS within amino acids 1 to 75 of Rps2. However, none of the fragments was completely shifted to the cytoplasm in the *pse1-1* mutant; therefore, other importins have to contribute to some extent to Rps2(1–145) nuclear import, at least in the absence of Pse1.

Next, we performed TurboID experiments to identify proteins in close proximity to Rps2(1–145) and Rps2(1–145).R_95_R_97_K_99_>A (Figure 4B, Appendix A, Appendix A). In both cases, as expected due to the presence of the N-terminal Tsr4-binding region, Tsr4 was strongly enriched in the affinity purification of biotinylated proteins. Pse1 was detected as well, although it was much less enriched than upon expression of TurboID-tagged Rps2(76–145) (Figure 3D). Moreover, the R_95_R_97_K_99_>A exchanges only had a minor effect on the extent of Pse1 enrichment in the context of Rps2(1–145) (Figure 4B). TurboID with full-length Rps2 (wild-type or containing the R_95_R_97_K_99_>A exchanges) yielded a similar extent of Tsr4 enrichment as observed in the case of Rps2(1–145) wild-type and R_95_R_97_K_99_>A mutant protein, while an enrichment of Pse1 could not be observed, potentially due to the short duration of the Rps2–Pse1 interaction compared to interactions of Rps2 in the context of the ribosome (Appendix A, Appendix A). To further characterize the effects of the R_95_R_97_K_99_>A exchanges on the interaction of the different Rps2 fragments with Pse1, we performed yeast two-hybrid (Y2H) analyses (Figure 4C, Appendix A). No interaction of any of the Rps2 variants was detected with full-length Pse1 (Appendix A), which was not surprising as importin–cargo interactions are generally only very short-lived in the nucleus due to the fact that the binding of RanGTP to the N-terminal arch of importins mediates cargo release [6]. To prevent cargo dissociation and hence enable productive importin–cargo Y2H interactions in the nucleus, we generated a Pse1 variant with a partial deletion of its N-terminal RanGTP-binding surface (Pse1.302C; starting with amino acid 302) [27]. As anticipated, utilization of the Pse1.302C variant permitted the detection of Y2H interactions between Pse1 and Rps2, Rps2(1–145), and Rps2(76–145) (Figure 4C, Appendix A). While the R_95_R_97_K_99_>A exchanges almost completely abolished the interaction of Rps2(76–145) with Pse1.302C, they only reduced the interaction of both Rps2 and Rps2(1–145) with Pse1.302C (Figure 4C, Appendix A).

Taken together, the above results indicate that the mutant Rps2(1–145) R_95_R_97_K_99_>A protein is still capable, albeit less efficiently than the wild-type counterpart, of interacting with Pse1 and can thus still be imported into the nucleus via Pse1. On the other hand, the mutant Rps2(76–145).R_95_R_97_K_99_>A protein interacts with Pse1 only poorly and its nuclear import is strongly impaired. We conclude that amino acids 1–75 of Rps2 must harbor a second import signal, which could also be recognized by Pse1.

### 3.5. Tsr4 Is Not Required for Import Mediated by the N-Terminal Rps2 Region

Considering that Tsr4 binds to the N-terminal region of Rps2 (Figure 2A, Ref. [7]), we speculated that Tsr4 might be involved in this second Rps2 import mechanism. To address this possibility, we examined the localization of Rps2(1–145).R_95_R_97_K_99_>A-3xyEGFP in the absence of Tsr4. This analysis is complicated by the fact that Tsr4 is an essential protein; however, in our previous study we found that *tsr4*∆ cells are viable, although displaying a severe slow-growth phenotype, when *RPS2* is provided on a low-copy number plasmid in a *rps2*∆ strain, presumably resulting in an increased *RPS2* copy number compared to the single-copy presence of *RPS2* in a wild-type strain [7]. Building on this knowledge, we transformed the 3xyEGFP reporter plasmids into a *tsr4*∆ *rps2*∆ strain complemented by a *URA3*-*RPS2* plasmid and, as a control, into a Tsr4-expressing *rps2*∆ *URA3*-*RPS2* strain. As previously observed, the Rps2-3xyEGFP reporter accumulated in the nucleus in cells lacks Tsr4 (*tsr4*∆ *rps2*∆ [*RPS2*] strain) (Figure 5, Appendix A), suggesting that the nuclear import of Rps2 can occur in the absence of Tsr4, and that, moreover, efficient Rps2 incorporation into pre-ribosomes is dependent on Tsr4 [7].

Notably, a strong nuclear accumulation was also observed for Rps2.R_95_R_97_K_99_>A-3xyEGFP in the absence of Tsr4, while Rps2(1–145)-3xyEGFP and Rps2(1–145).R_95_R_97_K_99_>A-3xyEGFP localized to the nucleus, mostly within intense dot-like structures outside the nucleolus that could correspond to aggregates, both in cells containing or lacking Tsr4 (Figure 5). We conclude that the second import mechanism also utilized by Rps2, involving amino acids 1–75, does not depend on Tsr4.

### 3.6. The N-Terminal Rps2 NLS Overlaps with the Tsr4-Binding Region

The partial nuclear localization of the Rps2(1–42)-3xyEGFP fusion protein (Figure 1B), together with the occurrence of an RG-rich sequence within the 28 N-terminal amino acids of Rps2 (Appendix A), which might potentially represent an RG-NLS, prompted us to test whether the very N-terminal region of Rps2 is required for the nuclear import of Rps2(1–145).R_95_R_97_K_99_>A-3xyEGFP. Notably, Tsr4 binds approximately to the same region, as suggested by our previous study in which we mapped the Tsr4-binding site to amino acids 1–42 of Rps2 [7]. We reasoned that it might be possible to map the Tsr4-binding site to an even shorter Rps2 fragment by generating further N- and C-terminal truncation variants and testing their capacity to interact with Tsr4 in Y2H assays (Figure 6A, Appendix A). Indeed, Rps2 missing the N-terminal five or ten amino acids still showed full interaction with Tsr4. Moreover, the 33 or 28 N-terminal residues alone were sufficient to confer a robust Y2H interaction with Tsr4, while, as already previously described [7], Rps2(1–22) interacted only weakly with Tsr4. Finally, we combined the above N- and C-terminal truncations which supported full interaction, and found that all tested combinations still interacted equally well with Tsr4, with the shortest tested fragment displaying full interaction being Rps2(10–28) (Figure 6A, Appendix A).

Next, we wanted to address whether mutation of the Tsr4-binding site would hamper the putative N-terminal NLS. To this end, we generated constructs expressing 3xyEGFP fusions of either a variant lacking the 28 N-terminal amino acids of Rps2 (Rps2(29–145)) or a Rps2(1–145) fragment, termed Rps2(1–145).KR_10–28_>A, having all basic residues within amino acids 10 to 28 (one K and seven R residues; see Appendix A) exchanged to As. Moreover, both variants were generated with and without the R_95_R_97_K_99_>A exchanges. The variants containing only the N-terminal manipulations (Rps2(29–145)-3xyEGFP and Rps2(1–145).KR_10–28_>A-3xyEGFP) still showed a nuclear localization similar to (Rps2(1–145)-3xyEGFP, with a stronger signal in the nucleoplasm than in the nucleolus (Figure 6B, Appendix A). In contrast, both variants additionally carrying the exchanges affecting the internal Rps2 NLS (Rps2(29–145).R_95_R_97_K_99_>A-3xyEGFP and Rps2(1–145).KR_10–28_>A/R_95_R_97_K_99_>A-3xyEGFP) failed to accumulate in the nucleus. We conclude that besides the NLS within amino acids 76–145, to which R95, R97, and K99 make an essential contribution, Rps2 contains a second NLS in its N-terminal region, which overlaps with the Tsr4-binding site and critically depends on the presence of several basic residues within a short stretch ranging from amino acid 10 to 28. Notably, this sequence stretch is present in the Rps2(1–42)-3xyEGFP reporter fusion, which localizes to both the cytoplasm and nucleus (Figure 1B). We hypothesized that the binding of Tsr4 might affect the nuclear import of this fragment through the N-terminal NLS by potentially modulating the efficiency of importin binding. To test this hypothesis, we again utilized a *tsr4*∆ *rps2*∆ strain complemented by a *URA3*-*RPS2* plasmid and, as a control, a *rps2*∆ [*URA3*-*RPS2*] strain. Both strains were transformed with the Rps2(1–42)-3xyEGFP reporter plasmid. In contrast to the wild-type strain, where the Rps2(1–42)-3xyEGFP signal was stronger in the nucleus compared to the cytoplasm (Figure 1B), the *rps2*∆ [*URA3*-*RPS2*] strain displayed an even distribution of Rps2(1–42)-3xyEGFP between the cytoplasm and the nucleus (Figure 6C). Interestingly, the *tsr4*∆ *rps2*∆ [*URA3*-*RPS2*] strain, lacking Tsr4, exhibited a slight accumulation of the Rps2(1–42)-3xyEGFP reporter fusion in the nucleus, suggesting that nuclear import via the N-terminal NLS of Rps2 might be more efficient in the absence of Tsr4.

## 4. Discussion

With this study, we have provided insights into the intricate mechanisms underlying nuclear import of the r-protein Rps2. We found that amino acids 76 to 145 are sufficient to target the protein to the nucleus, with residues R95, R97, and K99 being essential for the nuclear localization of this fragment. The main importin responsible for import via Rps2(76–145) is Pse1. Hence, the preference of Rps2 for Pse1 deviates from the common preference of r-proteins for Kap123, with Pse1 stepping in place mainly in the absence of Kap123 [13]. Our data moreover demonstrate that the mutation of R95, R97, and K99 in the Rps2(76–145) fragment greatly reduces the interaction with Pse1, suggesting that these residues are critical determinants for Pse1 binding. Previous structural analyses of Pse1 in complex with NLS sequences of three different Pse1 cargoes have led to the definition of the IK-NLS with the consensus K-V/I-X-K-X_1–2_-K/H/R [27,28]. The segment ranging from residues 95 to 99 of Rps2 (RTRFK), however, does not match this consensus. Moreover, it is positioned within a beta-sheet (Appendix A), while IK-NLSs are unstructured [27,28]. Hence, Rps2 likely uses a binding mode that differs from the one reported for the interaction of Pse1 with IK-NLSs, and seems to involve structured elements. In our tandem affinity purification experiment, where we expressed either Rps2(76–145)-TAP or Rps2(76–145).R_95_R_97_K_99_>A-TAP from plasmids in a wild-type strain, we observed higher levels of the Rps2 fragment carrying the substitutions in cell lysates compared to the wild-type fragment (Figure 3C). This observation suggests that the substitutions of R95, R97, and K99 to A might induce structural changes that enhance the stability of the Rps2(76–145) fragment. These altered structural features might impede the efficient binding of Pse1, despite promoting protein stability.

The Rps2(1–145) fragment, containing in addition to the above-discussed nuclear-targeting domain also the N-terminal part of Rps2, enters the nucleus as well, even when the residues critical for the nuclear targeting of the Rps2(76–145) fragment are mutated. Moreover, R95A, R97A, and K99A mutation reduces the Y2H interaction of the Rps2(1–145) fragment with Pse1 only slightly, while the interaction of Rps2(76–145) with Pse1 is severely reduced by these exchanges. This suggests that Pse1 may possess an additional binding site within amino acids 1–75 of Rps2. Indeed, Rps2(1–145).R_95_R_97_K_99_>A-3xEGFP displayed an increased cytoplasmic signal in *pse1-1* mutant cells compared to wild-type cells, indicating that even if the internal NLS is not available for interaction with Pse1, Pse1 is capable of importing the Rps2(1–145) fragment. It is worth noting that none of the tested Rps2 fragment 3xEGFP fusions showed complete import inhibition in the *pse1-1* mutant, implying that, as also suggested in previous studies (see for example [29,30,31]), other importins can compensate for the loss of one importin. Nevertheless, the significant defects observed in the *pse1-1* mutant strongly indicate that Pse1 is the primary importin binding to the two Rps2 NLS elements described in this study.

Nuclear import via this N-terminal nuclear-targeting region requires basic residues within the RG-rich, unstructured N-terminal part (amino acids 10–28) of Rps2. It is already known that such RGG regions can function as NLSs for Kap104 [32,33]; however, recognition of RG-rich NLSs by Pse1 has not been reported so far. Although our findings suggest that Rps2(1–145) can still be imported into the nucleus by Pse1 when either the N-terminal or the internal NLS is mutated, it remains unclear whether Pse1 interacts simultaneously with both binding sites in the wild-type scenario, or if it only utilizes one of them at a time. It will be interesting to further define and map the two Pse1-binding regions of Rps2 in the future, which might lead to the definition of novel NLS consensus motifs for Pse1. Notably, while the N-terminal and internal NLSs share some sequence similarities, such as positively charged amino acids with similar spacing (e.g., 95-RTRFK-99 and 17-RNRGR-21), they are embedded in entirely different structural contexts. The N-terminal NLS resides within an unstructured region, whereas the internal NLS lies within a beta-sheet (Appendix A). Consequently, the two NLSs may employ distinct binding modes for Pse1 interaction.

It is important to acknowledge that the basic residues within amino acids 10–28 of Rps2, although being necessary for the nuclear targeting of a Rps2(1–145) fragment with a mutated internal NLS (Figure 6B), are not sufficient for efficient import, as concluded from the fact that a small Rps2 fragment containing these amino acids, Rps2(1–42), does not exclusively localize to the nucleus (Figure 1B and Figure 6C). Hence, additional sequence elements are required for the complete functionality of the N-terminal NLS. Furthermore, it is possible that not all eight positively charged amino acids within Rps2(10–28) are essential for the function of the N-terminal NLS. It is plausible that a few specific residues within this range are crucial for the import via the N-terminal NLS, or that multiple clusters of positively charged amino acids within this sequence can fulfill this function alternatively, as recently reported for the NLS of the viral protein HIV-1 Tat [34]. Future in-depth biochemical and structural studies might provide further insights into the binding modes and interplay between the two Rps2 NLSs.

Importantly, the amino acids required for the function of the N-terminal NLS of Rps2 overlap with the binding site of its dedicated chaperone, Tsr4. Therefore, it was important to investigate whether the presence of Tsr4 affects the nuclear targeting of Rps2 via the N-terminal NLS.

We can exclude the possibility that import mediated by Rps2 amino acids 10–28 occurs via a ‘piggyback’ mechanism in which Tsr4 binds Rps2 and provides the NLS for the nuclear import of the Rps2-Tsr4 complex, as our data revealed that the presence of Tsr4 is not required for the import involving the N-terminal region of Rps2 (Figure 5). Rps2(1–42)-3xyEGFP even exhibited a stronger nuclear signal in the absence of Tsr4 (Figure 6C), suggesting that its import is less efficient when Tsr4 is present. One potential explanation for this effect is that Tsr4 shields the N-terminal NLS, thereby reducing the efficiency of importin binding. It is yet to be determined whether Tsr4 accompanies Rps2 into the nucleus or dissociates from Rps2 already in the cytoplasm, e.g., upon the binding of importin. Tsr4-GFP does not accumulate in the nucleus upon inhibition of the main exportin Crm1 [10]. However, the human Tsr4 homolog PDCD2 accompanies human RPS2 into the nucleus [35], as does the closely related PDCD2L [36]. Further, our data demonstrate that in the absence of Tsr4, Rps2 accumulates in the nucleolus (Figure 5 and [7]), suggesting that Rps2’s efficient incorporation into pre-ribosomal particles is prevented. The simplest explanation for this phenotype would be that Tsr4 functions in promoting Rps2 pre-ribosome incorporation in the nucleus. On the other hand, the more efficient import of the Rps2(1–42) fragment in the absence of Tsr4 suggests that nuclear import might occur after the release of Tsr4. Alternatively, the interaction of Pse1 with the internal NLS of Rps2 may be sufficient to mediate the nuclear targeting of Rps2 bound to Tsr4, even if the N-terminal NLS is not fully accessible to the importin.

The binding of Pse1 to Rps2 could serve a second function beside nuclear import as importins have been implicated in functioning as chaperones for exposed basic domains [12]. The richness in positive charges, together with the flexibility of the Rps2 N-terminal region, might make Rps2 particularly prone to aggregation, which could be the reason why two different, potentially redundant mechanisms for chaperoning of this region have evolved, with the main one relying on a dedicated chaperone and the second one involving an importin.

Interestingly, the N-terminal RG-rich region of Rps2 is absent in bacteria and archaea (Appendix A), suggesting that it serves a eukaryote-specific function, as is the case for a targeting sequence for nuclear import. In contrast, parts of the internal positively charged NLS residues are also found in archaea and bacteria. For instance, *Pyrococcus furiosus* uS5 contains all three of these residues, while *Sulfolobus* and *Bacillus subtilis* have two positively charged amino acids in the corresponding region (Appendix A). NLS-type motifs have been observed in archaea before, suggesting that NLS sequences may have originated from sequences that originally served other functions [37,38].

Intriguingly, Rps2’s unstructured N-terminal region seems to be a hub for the binding of multiple interaction partners (elaborated in detail in a review article by the Bachand group within this Special Issue [39]). Besides the binding partners investigated in this study (Tsr4 and importins), the N-terminal part of Rps2 also likely interacts (at least transiently) with Hmt1, as this enzyme methylates an arginine in the N-terminal region of Rps2 [40,41]. In the human system, RPS2 is bound by PDCD2 or PDCD2L, and is additionally stably bound by the arginine methyl transferase PRMT3, which competes with the zinc finger protein ZNF277 for RPS2 binding [22,36,42,43]. The investigation of the timing and coordination of these manifold interactions will be an interesting subject for future studies.

## Figures and Tables

**Figure 1 biomolecules-13-01127-f001:**
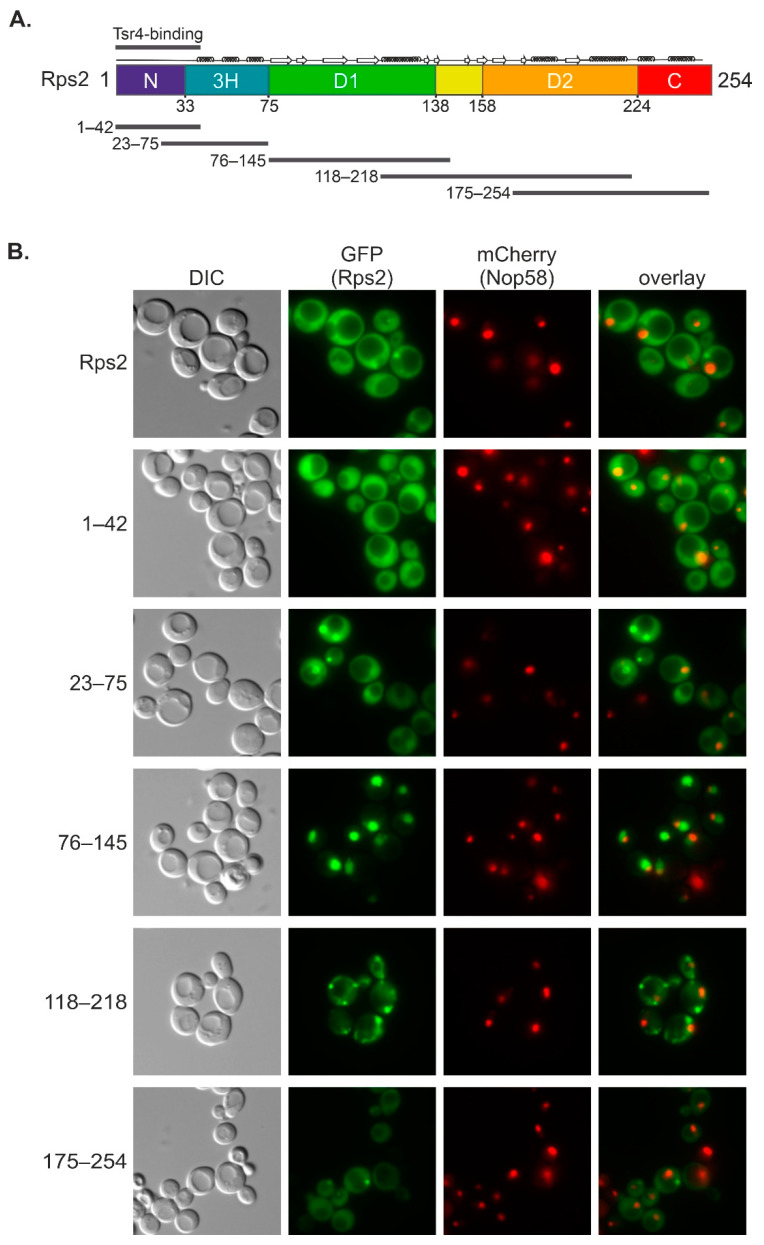
Rps2 residues 76–145 are sufficient to target a 3xyEGFP reporter to the nucleus. (**A**) Schematic representation of Rps2 with secondary structure elements and overview of fragments tested in (**B**). Indicated domains according to the Rps2 structure shown in Appendix A are as follows: N, eukaryote-specific N-terminal extension; 3H, three-helix element; D1, domain one; D2, domain two; C, C-terminal extension. The Tsr4-binding site, as previously determined [7], is indicated on top of the schematic representation. (**B**) Fluorescence microscopy of a strain expressing Nop58-yEmCherry (nucleolar marker) as well as 3xyEGFP fusions of Rps2 or the indicated truncated Rps2 fragments. DIC, differential interference contrast.

**Figure 2 biomolecules-13-01127-f002:**
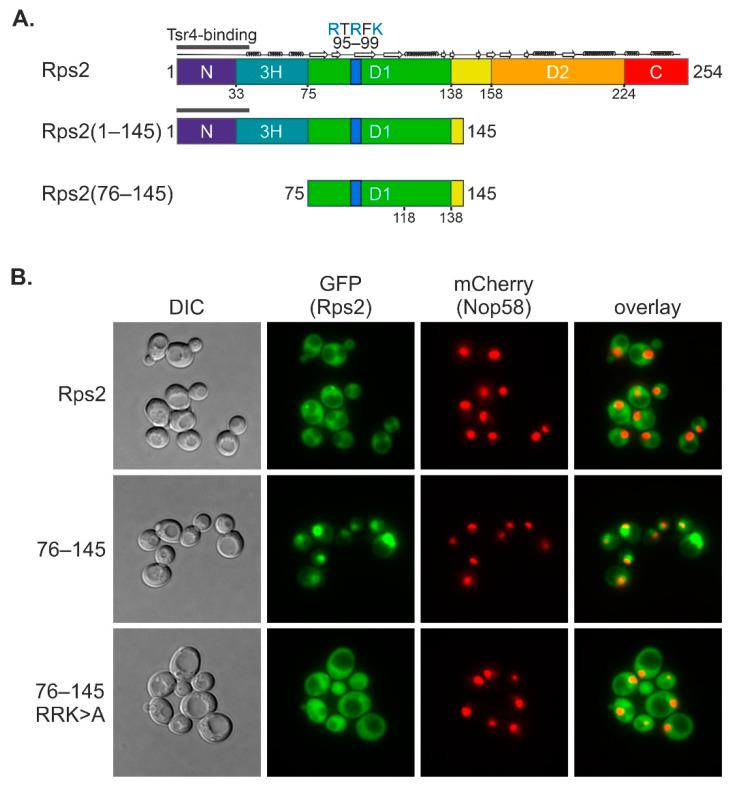
Residues R95, R97, and K99 are essential for nuclear targeting of Rps2(76–145). (**A**) Overview of the main Rps2 fragments tested in this study. (**B**) Fluorescence microscopy of a strain expressing Nop58-yEmCherry as well as Rps2-3xyEGFP or Rps2(76–145)-3xyEGFP with or without exchanges of the three basic residues (R_95_R_97_K_99>A_, abbreviated RRK>A), comprised in the 95-RTRFK-99 stretch that are part of the putative NLS. In this experiment, the intensities of the GFP fluorescence signals were adjusted for better comparison. The original, identically processed pictures of the adjusted images are shown in Appendix A.

**Figure 3 biomolecules-13-01127-f003:**
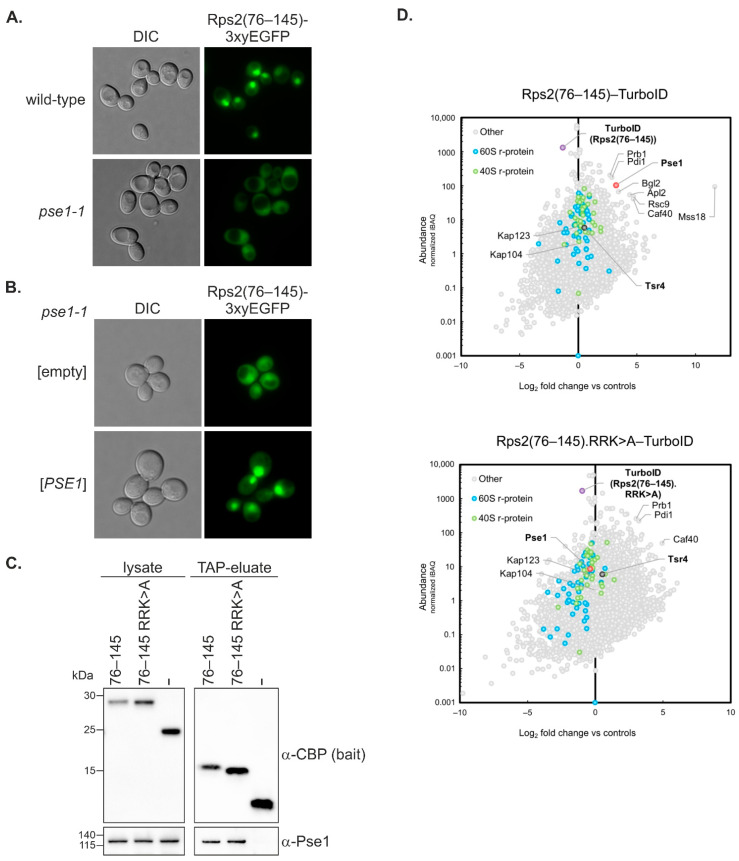
Pse1 mediates nuclear import of Rps2(76–145). (**A**) Localization of Rps2(76–145)-3xyEGFP visualized by fluorescence microscopy in the wild-type strain and the importin mutant strain *pse1-1*. The localization of the fusion protein in additional importin mutant strains is shown in Appendix A. (**B**) Complementation assay. The *pse1-1* strain was transformed with a *PSE1*-harboring *URA3* plasmid or the empty control plasmid, as well as with the Rps2(76–145)-3xyEGFP *LEU2* reporter plasmid, and transformants were inspected by fluorescence microscopy. Growth assays of the same strains are shown in Appendix A. (**C**) Pse1 co-purifies with Rps2 in vivo. Rps2(76–145)-TAP with and without the R_95_R_97_K_99>A_ exchanges, as well as the TAP tag alone as negative control (-), were expressed from plasmids in a wild-type strain. After TAP purification, lysates and TAP eluates were analyzed by Western blotting using α-CBP and α-Pse1 antibodies. (**D**) TurboID-based proximity labeling with Rps2(76–145) and Rps2(76–145).R_95_R_97_K_99>A_ as baits. The normalized abundance value (iBAQ, intensity-based absolute quantification; y-axis) of each protein detected in the respective purification is plotted against its relative abundance (log_2_-transformed enrichment; x-axis). Relative abundance was calculated compared to the averaged protein abundance in the two control purifications (derived from cells individually expressing the GFP-TurboID and the NLS-GFP-TurboID bait, which accounts for the cytoplasmic and nuclear background, respectively). Proteins that are enriched compared to the negative controls can be found on the right side of the Christmas tree plot. The names of proteins that are particularly enriched, as well as importins Pse1, Kap123, and Kap104 are indicated. The bait proteins and Pse1 are highlighted by bold letters.

**Figure 4 biomolecules-13-01127-f004:**
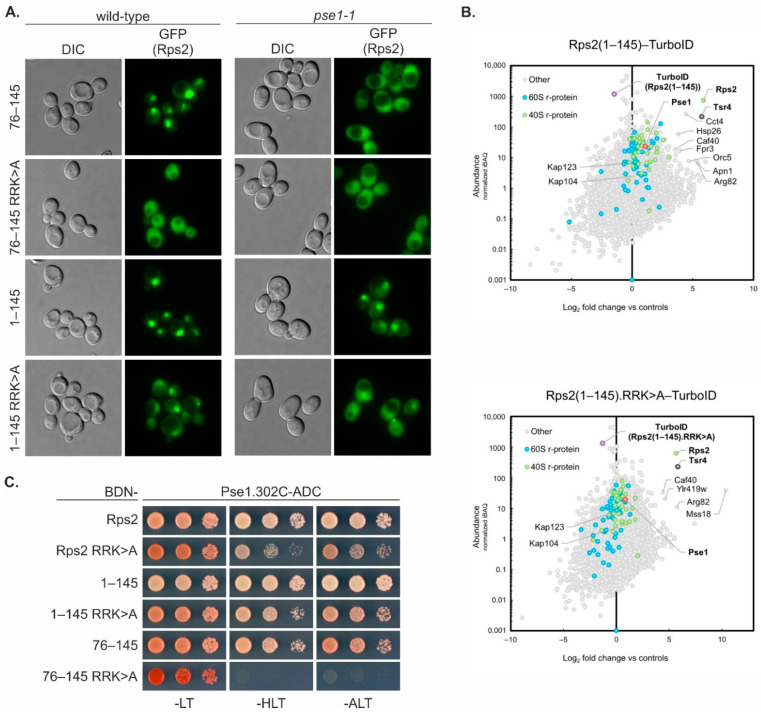
Rps2 contains a second NLS in its N-terminal extension. (**A**) Fluorescence microscopy of a wild-type as well as a *pse1-1* mutant strain expressing 3xyEGFP fusions of the indicated wild-type or mutated Rps2 fragments. In this experiment, the intensities of the GFP fluorescence signals were adjusted for better comparison. The original, identically processed pictures of the adjusted images are shown in Appendix A. (**B**) TurboID-based proximity labeling with Rps2(1–145) and Rps2(1–145).R_95_R_97_K_99_>A as baits. The normalized abundance value (iBAQ) of each protein detected in the respective purification is plotted against its relative abundance (log_2_-transformed enrichment) compared to the averaged abundance in the control purifications (GFP-TurboID and NLS-GFP-TurboID). The names of proteins that are particularly enriched, as well as importins Pse1, Kap123, and Kap104, are indicated. The bait proteins, Tsr4, and Pse1 are highlighted by bold letters. (**C**) Yeast two-hybrid (Y2H) interaction assay between Pse1 lacking the 301 N-terminal amino acids (Pse1.302C), C-terminally fused to the Gal4 activation domain (AD), and Rps2 and the indicated fragments thereof (including, when indicated, the RRK>A exchanges) containing the Gal4 DNA-binding domain (BD) at the N-terminal end. Growth on SDC-his-leu-trp plates (labeled -HLT) indicates a weak interaction; growth on SDC-ade-leu-trp plates (labeled -ALT) indicates a strong Y2H interaction. SDC-leu-trp (labeled -LT) served as growth control. For Y2H assays with the same Rps2 proteins containing the Gal4 DNA-binding domain at the C-terminal end, as well as the Y2H assays between the N- or C-terminally-fused Rps2 variants and full-length Pse1 or the Gal4 activation domain alone (negative control), see Appendix A.

**Figure 5 biomolecules-13-01127-f005:**
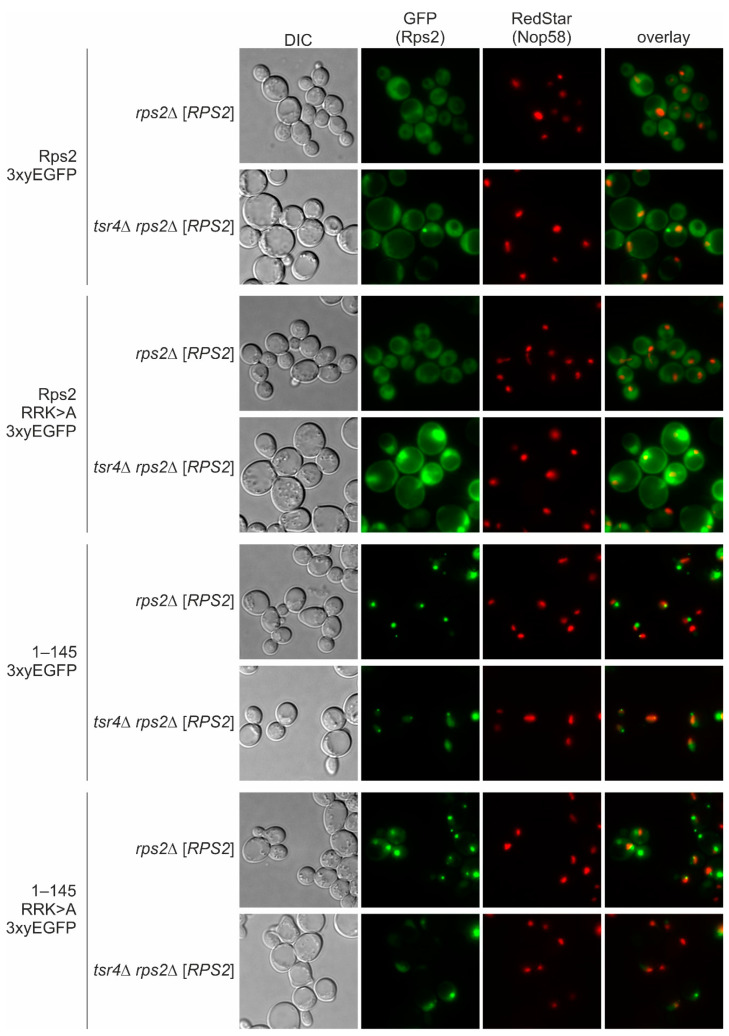
Tsr4 is not required for import mediated by the N-terminal Rps2 region. Fluorescence microscopy of *rps2*Δ and *tsr4*Δ *rps2*Δ strains, containing a *URA3*-*RPS2* plasmid, expressing the indicated Rps2-3xyEGFP fusion proteins from *LEU2* plasmids, and a chromosomal C-terminal RedStar2 fusion of Nop58. Each panel was processed individually to make the observed phenotypes more apparent. To allow for the evaluation of the differences in signal intensities, the same panels, but all identically processed, are shown in Appendix A.

**Figure 6 biomolecules-13-01127-f006:**
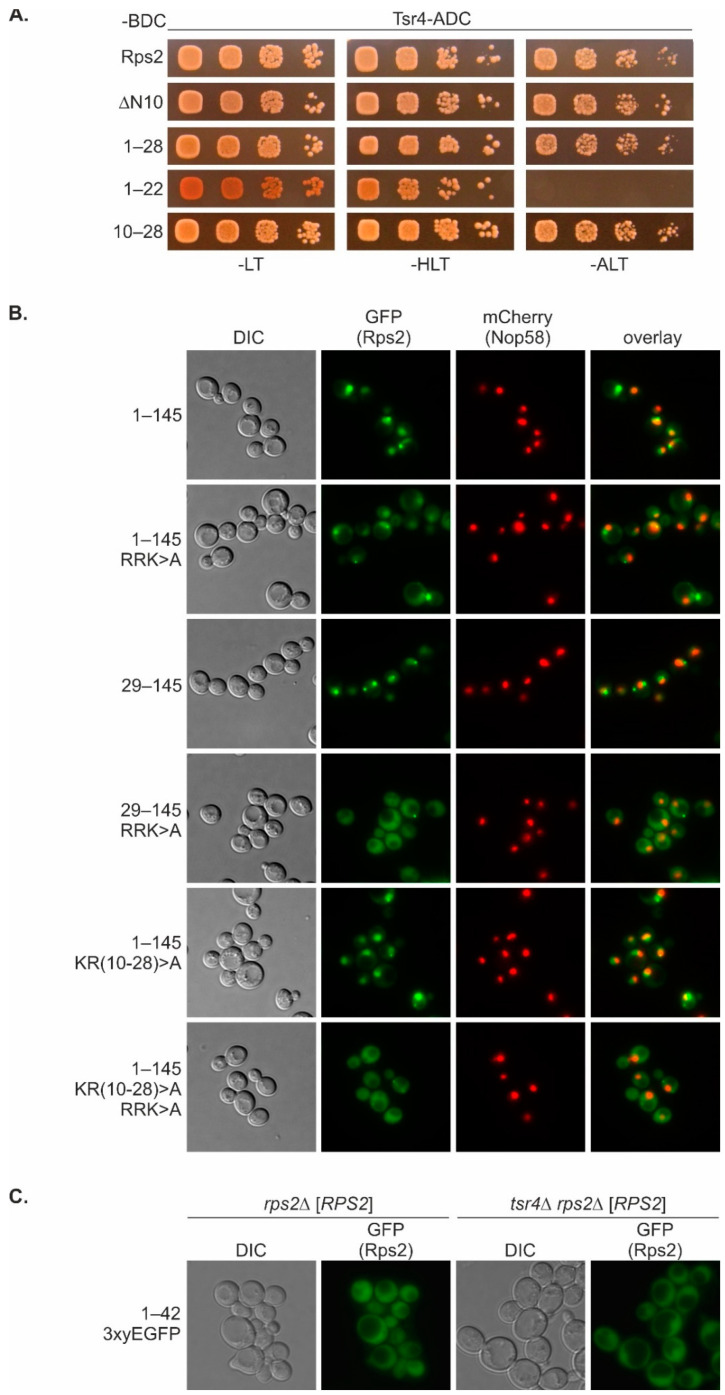
The N-terminal Rps2 NLS overlaps with the Tsr4-binding region. (**A**) Yeast two-hybrid (Y2H) assays between full-length Tsr4, C-terminally fused to the Gal4 activation domain (AD), and Rps2 and fragments thereof, and C-terminally fused to the Gal4 DNA-binding domain (BD). For more details, see the legend of Figure 4C. For results with additional fragments as well as negative controls, see Appendix A. (**B**) Fluorescence microscopy of a strain expressing Nop58-yEmCherry as well as 3xyEGFP fusions of the indicated wild-type or mutated Rps2 fragments. In this experiment, the intensities of the GFP fluorescence signals were adjusted for better comparison. The original, identically processed pictures of the adjusted images are shown in Appendix A. (**C**) Fluorescence microscopy of *rps2*Δ and *tsr4*Δ *rps2*Δ strains, containing a *URA3*-*RPS2* plasmid, expressing Rps2(1–42)-3xyEGFP from an *LEU2* plasmid.

## Data Availability

TurboID results are provided in Appendix A.

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
