# Peer review of "Dissecting the Nuclear Import of the Ribosomal Protein Rps2 (uS5)"

_biomolecules, 2023, doi:10.3390/biom13071127_

Round 1

Author Response

Dear reviewer 1, thank you for your constructive suggestions, which helped us to further improve our manuscript. Please find below a point-by-point response to your comments.

Here are a few major concerns.

  1. Visual inspection of the cell images shows contrasting trends to what is mentioned in the text. Quantification of the fluorescence in the cell images is essential to properly understand and interpret data. 1.B. {e.g.: a slight shift of Rps2(76-145)-3xyEGFP localization to the cytoplasm was detected in the kap123Δ strain (Figure 3A). Notably, a strong reduction in nuclear accumulation of Rps2(76-145)- 3xyE-GFP was observed in pse1-1 single mutant and pse1-1 kap123Δ double mutant cells (Figure 3A)}.

Response: We thank referee 1 for this comment. We re-inspected the figures in the file that was sent for review and noticed that in the course of PDF conversion, the image quality has suffered. Instead of submitting a PDF file, we are submitting a word document this time, in which we directly inserted the images. Now the images are of better quality, and we are convinced that all the effects described in the text are also visible in the Figures. We did not perform quantifications as this would require to assign the region corresponding to the nucleus in the image of each cell to be quantified. As the used marker protein Nop58 is a reporter protein of the nucleolus, which only represents a sub-compartment of the nucleus, it is not possible to use this marker to define the region corresponding to the nucleus. Hence, it would be necessary to manually assign the nucleus in each cell, which would be particularly difficult for cells which show a similar signal intensity in the nucleus and cytoplasm. However, we made some improvements that will help to better interpret our results, i.e. we added some extra controls and moreover made sure that all major findings are supported in the manuscript by more than one microscopy picture. E.g. the localization of Rps2(76-145)-3xyEGFP is shown in Figure 1B, Figure 2B, and Figure 4A, the localization of Rps2(76-145).R95R97K99>A-3xyEGFP is shown in Figure 2B and Figure 4A, the localization of Rps2(1-145)-3xyEGFP and Rps2(1-145).R95R97K99>A-3xyEGFP is shown in Figure 4A and Figure 6B.

We apologize for the data on kap123Δ in our initial submission. The reduction of nuclear signal of Rps2(76-145)-3xyEGFP we described for the kap123Δ strain occurred only in some cells, while Rps2(76-145)-3xyEGFP still showed a strong nuclear localization in other cells. To find out whether the localization of the 3xyEGFP fusion protein is truly affected by the absence of Kap123, we transformed the kap123Δ strain with a plasmid carrying the KAP123 wild-type sequence and tested if this would result in a complementation of the phenotype (new Supplementary Figure 3B and 3C). As the relative distribution of the reporter fusion protein was not altered in the presence of KAP123, we conclude that contrary to our original interpretation, kap123Δ cells do NOT show a significant nuclear import defect of the reporter construct and that our interpretation of a slight shift to the cytoplasm due to KAP123 deletion was a misinterpretation. We sincerely apologize for this mistake and now corrected the text accordingly.

In contrast, the findings on the effects in the pse1-1 mutant are very clear, consistently showing a reduction of the nuclear compared to the cytoplasmic Rps2(76-145)-3xyEGFP signal in the pse1-1 mutant. This is also nicely underscored by the complementation of this import defect upon transformation with a plasmid containing the PSE1 wild-type sequence. We previously showed that result in the supplementary data, but we now moved it to Figure 3B to strengthen our claim. Moreover, we performed a new experiment, in which we compared the localization of Rps2(76-145)-3xyEGFP with the localization of Rps2(1-145)-3xyEGFP, as well as of the corresponding RRK>A mutant variants of the fragments in the pse1-1 mutant (Figure 4A of the revised manuscript). Also, these results further support our conclusions on the reduced import of Rps2(76-145)-3xyEGFP import in the pse1-1 strain, and moreover provide evidence that Pse1 is also involved in the import of Rps2(1-145)-3xyEGFP via the N-terminal NLS.

  1. Line 358-360 2. Is it the changes in binding affinity of Pse1 to mutant Rps2 that causes decrease in Pse1- dependent export?

Response: Indeed, our results suggest reduced binding of Pse1 to mutant Rps2 (Figure 3C and Figure 3D). This reduced binding is most likely the cause of the decreased import of Rps2(76-145).R95R97K99>A -3xyEGFP. In the revised version of this manuscript, we provide a better explanation for these interpretations in the results text (lines 436-440).

  1. How confident the authors are that the structures of Rps2 fragments are native like? If the structure of Rps2 is critical for the recognition Pse1, structural integrity of the fragments is essential.

Response: For the experiment in Figure 1B, we designed the fragments based on the domain organization of Rps2. Nevertheless, we cannot exclude that in reality, some of these fragments, particularly those with aberrant localizations (e.g. the 118-218 fragment which displayed a mitochondrial localization), might be misfolded. However, it has to be noted that experiment 1B is a screen aiming at identifying a fragment with nuclear localization. The main conclusion from the experiment is that fragment 76-145 contains a NLS and is most likely in the correct conformation, otherwise it would not localize to the nucleus.

In the case of the Rps2(76-145) fragment with the R95, R97, and K99 to A exchanges, it is possible that the amino acid exchanges result in local changes in the folding. Most likely, the exchanges result in a stabilization of the protein, considering that we detected higher protein levels of Rps2(76-145).R95R97K99>A-TAP compared to the wild-type variant Rps2(76-145)-TAP in the cells (Figure 3C). We agree that it is likely that not only the sequence, but also the structure of the NLS is relevant for its function. We discuss these possibilities in the discussion section of the revised manuscript (lines 629-636).

  1. Why is the overlap of Nop10 and Rps2 signals are not perfect in many instances where authors suggest nuclear localization?

Response: Nop58 localizes exclusively to the nucleolus, but is not found in the rest of the nucleus (the nucleoplasm). The reason why we decided to use a nucleolar marker is that the nucleolus is the site where ribosome biogenesis starts, and where most ribosomal proteins are incorporated into pre-ribosomal particles. Therefore, we were interested to find out if fragments that are capable of entering the nucleus are targeted to the nucleolus or not. We apologize for not having provided this information. In the revised manuscript, we are now explaining the rationale for selection of Nop58-yEmCherry as a marker (lines 292-299). Moreover, for all the reporter fusions for which we detected a nuclear localization, we now provide details where in the nucleus the signal was observed. Actually, the Rps2(76-145)-3xyEGFP as well as the Rps2(1-145)-3xyEGFP reporter fusions both localized to the entire nucleus (with an even stronger signal in the region not overlapping with Nop58), suggesting that they contain an NLS but lack elements for targeting to the nucleolus.

  1. Minor concerns 1. Had a very difficult time understanding the figures (especially Figure 3 and 4). Axes of TurboIDbased proximity assay graphs should be labeled clearly, and figure legends need to be improved to increase the ease of reading.

Response: We apologize for not being clear in the figure labeling and legends. In the revised manuscript, we improved figure labels and legends. In particular, we simplified the labels for the TurboID data.

Reviewer 2 Report

I find the article submitted for review extremely interesting and important. However, I have some serious comments on the experiments presented, as well as on their interpretation. 

1. The authors presented the following interpretation of their results.

"In the presence of Tsr4, the unstructured N-terminal region of Rps2 is co-translationally captured by Tsr4, shielding the N-terminal NLS; in this scenario, Pse1 would not have access to the N-terminal NLS, but would exclusively use the internal NLS of Rps2. In conditions of limited amounts of Tsr4, the N-terminal NLS would become accessible for Pse1, allowing import also via this region".

The authors showed that in the case of Rps2(1-145).R95R97K99>A accumulation occurs in the nucleus. But I could not find data on whether NLS in the N-terminal site works only in the case of inactivation of the internal NLS. The authors use data from TurboID experiments, but this data seems insufficient. Two proteins can bind to the same region (and importin can also function as a chaperone, i.e., Tsr4 displacement should not lead to any abrupt aggregation).

The contribution of each of NLS was practically not evaluated. I think it is necessary to put up an experiment comparing localization of Rps2(1-145), Rps2(76-145) and Rps2(1-145).R95R97K99>A in pse1-1 single mutant cells. This will allow us to show more precisely the role of Pse1 in N-terminal NLS activity.

2. I have doubts about the data on the role of Kap123 in the accumulation of Rps2 in the nucleus (Fig. 3A). The deletion effect of kap123 seems insignificant and without quantification the conclusion does not look convincing. I would like to clarify this point.

3. The authors discuss the problem of NLS interaction with chaperone and importin at once. Recently, this feature of NLS has been linked to the origin of this type of sequence (doi: 10.1093/molbev/msz207). I believe the authors need to discuss their data in light of this paper as well (especially since it is also done on ribosomal proteins).

4. It is noteworthy that the N-terminal site is quite long and contains many positively charged amino acids. As I understood the authors' idea, they believe that this site must have some short binding site for Pse1 (lines 572-574). Indeed, 17RNRGR21 is similar to 95RTRFK99. However, it has recently been shown that long regions enriched with positively charged amino acids can be bound in different parts by importins (doi: 10.1128/JVI.01505-21). The authors of this article speculate that the same situation may be true for ribosomal proteins (they worked on the viral Tat protein). I think an accurate mapping of the NLS is necessary (using 3 additional mutants to different part of 10-28 region - 10,11 >A, 17,19,21 >A and 24,25,28>A).

Author Response

Dear reviewer 2, thank you for your constructive suggestions, which helped us to further improve our manuscript. Please find below a point-by-point response to your comments.

  1. The authors presented the following interpretation of their results.

"In the presence of Tsr4, the unstructured N-terminal region of Rps2 is co-translationally captured by Tsr4, shielding the N-terminal NLS; in this scenario, Pse1 would not have access to the N-terminal NLS, but would exclusively use the internal NLS of Rps2. In conditions of limited amounts of Tsr4, the N-terminal NLS would become accessible for Pse1, allowing import also via this region".

The authors showed that in the case of Rps2(1-145).R95R97K99>A accumulation occurs in the nucleus. But I could not find data on whether NLS in the N-terminal site works only in the case of inactivation of the internal NLS. The authors use data from TurboID experiments, but this data seems insufficient. Two proteins can bind to the same region (and importin can also function as a chaperone, i.e., Tsr4 displacement should not lead to any abrupt aggregation).

Response: The idea that Tsr4 and Pse1 compete for Rps2 binding to the N-terminal NLS is currently just one hypothesis, but we agree that alternative possibilities exist. In the revised version of the manuscript, we are now also discussing the possibility that Tsr4 and Pse1 bind to the N-terminal NLS at the same time (lines 706-710). Additionally, we performed an experiment to obtain further support for our hypothesis that Pse1 and Tsr4 might compete for binding to the Rps2 N-terminal NLS. Investigation of the localization of Rps2(1-42)-3xyEGFP (Figure 1B) had already suggested that this fragment alone (containing the stretch of positively charged residues) is not sufficient for efficient nuclear targeting. We reasoned that Tsr4 binding to that fragment might reduce the efficiency of importin binding. To address that, we investigated the localization of Rps2(1-42)-3xyEGFP in a Δtsr4 strain (new Figure 6C). Indeed, we observed a stronger nuclear signal of Rps2(1-42)-3xyEGFP in the strain lacking Tsr4 compared to the control strain, suggesting that the presence of Tsr4 might indeed reduce the efficiency of importin binding. However, also in the absence of Tsr4, we did not observe a full nuclear localization of Rps2(1-42)-3xyEGFP, suggesting that this fragment is not a good import substrate and that additional sequence or structural elements are required. We hope that we can gain more insights into these details in our future research.

The contribution of each of NLS was practically not evaluated. I think it is necessary to put up an experiment comparing localization of Rps2(1-145), Rps2(76-145) and Rps2(1-145).R95R97K99>A in pse1-1 single mutant cells. This will allow us to show more precisely the role of Pse1 in N-terminal NLS activity.

Response: We thank the reviewer for this excellent suggestion. We performed the suggested experiments (new Figure 4A). The signals of all tested reporter fusions shifted to the cytoplasm in the pse1-1 mutant, suggesting that Pse1 is also involved in import via the N-terminal NLS. Nevertheless, all fragments still showed a signal in the nucleus, in particular the Rps2(1-145)-3xyEGFP reporter fusion. We conclude that, although Pse1 seems to be the main importin, other importins can step in when Pse1 is mutated, which is to be expected as importins are known to function redundantly. We also discuss our conclusions on these results in the Discussion section of the new manuscript (lines 643-652).

  1. I have doubts about the data on the role of Kap123 in the accumulation of Rps2 in the nucleus (Fig. 3A). The deletion effect of kap123 seems insignificant and without quantification the conclusion does not look convincing. I would like to clarify this point.

Response: We apologize for this misinterpretation. Indeed, as also stated in our comment to reviewer 1, the reduction of nuclear signal was only observed in some of the cells, while other cells still showed a strong nuclear localization of the reporter fusion protein. As a control, we now included a complementation experiment, in which we transformed the strain with a plasmid carrying the KAP123 wild-type sequence and tested if this would lead to a complementation of the phenotype (new Supplementary Figure 3B and 3C). As the localization of the reporter fusion protein was identical irrespective of the presence or absence of KAP123, we conclude that our interpretation of a slight shift to the cytoplasm due to KAP123 deletion was indeed a misinterpretation. We corrected the text accordingly.

  1. The authors discuss the problem of NLS interaction with chaperone and importin at once. Recently, this feature of NLS has been linked to the origin of this type of sequence (doi: 10.1093/molbev/msz207). I believe the authors need to discuss their data in light of this paper as well (especially since it is also done on ribosomal proteins).

Response: Thank you for this suggestion, we are now discussing this paper (lines 725-733).

  1. It is noteworthy that the N-terminal site is quite long and contains many positively charged amino acids. As I understood the authors' idea, they believe that this site must have some short binding site for Pse1 (lines 572-574). Indeed,17RNRGR21 is similar to 95RTRFK99. However, it has recently been shown that long regions enriched with positively charged amino acids can be bound in different parts by importins (doi: 10.1128/JVI.01505-21). The authors of this article speculate that the same situation may be true for ribosomal proteins (they worked on the viral Tat protein). I think an accurate mapping of the NLS is necessary (using 3 additional mutants to different part of 10-28 region - 10,11 >A, 17,19,21 >A and 24,25,28>A).

Response: We agree that it would be very interesting to map this N-terminal NLS in more detail. We feel however that this goes beyond the scope of the present study. However, we plan to perform a fine-mapping of this NLS, as well as an investigation of the interplay between the two NLS elements in our future research. However, we included a discussion on the possibility mentioned by the reviewer that the long region containing positive charges could be bound in different parts by importins (lines 670-682).

Round 2

Reviewer 2 Report

The authors took into account all my suggestions, and it is a great pleasure for me to support the publication of this article.